# Red-light is an environmental effector for mutualism between begomovirus and its vector whitefly

Pingzhi Zhao[1,2☯], Xuan Zhang[1☯], Yuqing Gong[1,2☯], Duan Wang[1,2], Dongqing Xu[3], Ning Wang[1], Yanwei Sun[1], Lianbo Gao[1], Shu-Sheng Liu[4], Xing Wang Deng[5], Daniel J. Kliebenstein[6], Xueping Zhou[7], Rong-Xiang Fang[1,2], Jian Ye[1,2]*

1 State Key Laboratory of Plant Genomics, Institute of Microbiology, Chinese Academy of Sciences, Beijing, China, 2 CAS Center for Excellence in Biotic Interactions, University of Chinese Academy of Sciences, Beijing, China, 3 State Key Laboratory of Crop Genetics and Germplasm Enhancement, College of Agriculture, Nanjing Agricultural University, Nanjing, China, 4 Institute of Insect Sciences, Zhejiang University, Hangzhou, China, 5 State Key Laboratory of Protein and Plant Gene Research, Peking-Tsinghua Center for Life Sciences, School of Advanced Agriculture Sciences and School of Life Sciences, Peking University, Beijing, China, 6 Department of Plant Sciences, University of California, Davis, California, United States of America, 7 State Key Laboratory for Biology of Plant Diseases and Insect Pests, Institute of Plant Protection, Chinese Academy of Agricultural Sciences, Beijing, China

☯ These authors contributed equally to this work.
* jianye@im.ac.cn

**Data Availability Statement:** All relevant data are within the manuscript and its Supporting Information files or available from a public repository. Sequence data from this work can be

## Abstract

Environments such as light condition influence the spread of infectious diseases by affecting insect vector behavior. However, whether and how light affects the host defense which further affects insect preference and performance, remains unclear, nor has been demonstrated how pathogens co-adapt light condition to facilitate vector transmission. We previously showed that begomoviral βC1 inhibits MYC2-mediated jasmonate signaling to establish plant-dependent mutualism with its insect vector. Here we show red-light as an environmental catalyzer to promote mutualism of whitefly-begomovirus by stabilizing βC1, which interacts with PHYTOCHROME-INTERACTING FACTORS (PIFs) transcription factors. PIFs positively control plant defenses against whitefly by directly binding to the promoter of terpene synthase genes and promoting their transcription. Moreover, PIFs interact with MYC2 to integrate light and jasmonate signaling and regulate the transcription of terpene synthase genes. However, begomovirus encoded βC1 inhibits PIFs' and MYC2' transcriptional activity via disturbing their dimerization, thereby impairing plant defenses against whitefly-transmitted begomoviruses. Our results thus describe how a viral pathogen hijacks host external and internal signaling to enhance the mutualistic relationship with its insect vector.

## Author summary

Climate change is driving disease rapidly spread, esp. for global distribution of insect-borne diseases. This paper reports red-light as an environmental factor to promote insect vector olfactory orientation behavior and increase viral disease transmission. Plant virus

found in Genebank/EMBL or The Arabidopsis Information Resource (www.Arabidopsis.org) under the following accession numbers: AtPIF1 (AT2G20180), AtPIF3 (AT1G09530), AtPIF4 (AT2G43010), AtPIF5 (AT3G59060), AtMYC2 (At1G32640), AtTPS10 (At2g24210), AtTPS14 (AT1G61680), AtTPS21 (AT5G23960), TYLCCNV βC1 (AJ421621).Whiteflies were collected in the field in Chaoyang District, Beijing, China and were identified as Bemisia tabaci MEAM1, B biotype (mtCOI, GenBank accession number MF579701).

**Funding:** J.Y. was supported by the National Science Foundation of China (31830073, 31522046, 31672001) and National key research and development program (2019YFC1200503); P. Z. was supported by the National Science Foundation of China (31701783); X.Z. was supported by the Chinese Postdoctoral Science Foundation (2018M641509) and the National Science Foundation of China (31901853). The funders had no role in study design, data collection and analysis, decision to publish, or preparation of the manuscript.

**Competing interests:** I have read the journal's policy and the authors of this manuscript have the following competing interests: J.Y., Y.G. and P.Z. are inventors on a patent related to this work (China patent, 201810378111.4), filed by the Institute of Microbiology, Chinese Academy of Sciences.

adapts the supplemental red lighting practice in modern agricultural greenhouse production under protection, therefore enhancing disease spreading globally.

## Introduction

Climate change affects the emergence and spread of vector-borne infectious disease such as malaria, West Nile virus, Zika virus, and viral disease in staple crops via many ways [1,2]. Rising global temperatures can push disease-carrying insects such as mosquitoes and whiteflies to move into new places that affect the transmission of local viral pathogens [3]. Evidence suggests that crop production is threatened in complex ways by climate changes in the incidence of pests and pathogens [1,2]. Changed light condition also affects insect vector orientation and therefore feeding behavior. Arthropod-borne viruses (arboviruses) cause diseases in human and crops, and rely on their vectors for transmission and multiplication [4,5]. The distribution and population size of disease vectors can be heavily affected by local climate and light conditions. Besides of direct effecting fitness of their vectors, plant pathogens confer indirect effects on their vectors often by manipulating the plant defenses against the vector, e.g. volatile chemical components. These volatile substances act as olfactory clues, but also host-finding cues, defensive substances even sex pheromones [6,7]. Many of insect-borne plant pathogens, e.g. arboviruses of the families *Geminiviridae*, are capable of achieving indirect mutualistic relationships with vectors via their shared host plant [8–10].

To cope with these environmental changes, sessile plants have evolved integrated mechanisms to respond these complex stress conditions, minimizing damages, while conserving valuable resources for growth and reproduction [11–13]. As an energy source and a key environmental factor, light influences plant growth, defense, and even ecological structure [14,15]. The perception of light signals by phytochrome photoreceptors initiates downstream signaling pathways and regulates numerous plant processes during growth and defense [16]. The *Arabidopsis thaliana* PHYTOCHROME INTERACTING FACTORS (PIFs) are a class of basic helix-loop-helix (bHLH) transcription factors in *Arabidopsis*, which interact with the active photoreceptors to optimize plant growth and development [17–19].

Exogenous light signals integrated with endogenous signals from defense hormones such as jasmonate (JA) and salicylic acid in plant, mediate plant defense responses [14]. These defensive arsenals often produce a blend of ecologically important volatile chemicals such as terpenoids releasing to the environment, and counter the herbivore attack including vectors such as whitefly and aphid [14,20]. The downstream bHLH transcription factor MYC2 controls the production of some secondary metabolites, which can function as olfactory cues for insects, e.g. terpenoids and glucosinolates [21–24]. Although light is known to regulate plant growth and defense against insects and pathogens [25], the role of light as an environmental effector on the interaction of plant with pathogens and insects is not clarified.

Begomovirus, the largest genus of plant viruses and transmitted exclusively by whitefly, have evolved strategies to manipulate JA-regulated plant olfactory cues to promote their mutualism with whitefly vectors [9,22,26]. For example, the begomovirus Tomato yellow leaf curl China virus (TYLCCNV), which possesses only the DNA-A component with a betasatellite (TYLCCNB), is a whitefly-transmitted begomovirus that results in epidemic diseases in tomato, tobacco and other crops [27–29]. These host plants produce volatile terpenoids as olfactory repellents against whitefly [22,26,30]. We have previously shown that the TYLCCNB encoded a βC1 protein suppresses the transcriptional activation-activity of MYC2 by interfering with its dimerization, leading to reduced transcription of *TERPENE SYNTHASE* (*TPS*)

genes and terpenoid biosynthesis and anti-herbivory glucosinolates biosynthesis, thereby establishing an indirect mutualistic relationship between the pathogen and the vector [22]. Whether and how climate condition such as light affects this mutualism between begomovirus and whitefly herbivore has not been characterized.

Here, we report red-light as an environmental catalyzer to promote mutualism of whitefly-begomovirus. The family of multiple signaling integrator PIFs is a new key target of the viral βC1 protein. βC1 protein hijacks two kinds of bHLH transcription factors (MYC2 and PIFs) to decrease the transcription of *TPSs* genes that are expected to reduce terpene biosynthesis. Our results show that a begomovirus establishes an indirect mutualistic relationship with whitefly vector by modulating red light and JA signaling-mediated plant defense.

## Results

### Environment red-light is indispensable for betasatellite-encoded βC1 protein to promote host whitefly attraction

To detect whether light affects the natural begomoviral transmission process, we performed whitefly two-choice experiments using *Nicotiana benthamiana* (Nb) plants and Nb plants infected with TYLCCNV and its associated betasatellite (TYLCCNB) (TA+β) in white light and dark conditions (Fig 1A). Consistent with our previous report [22], whiteflies showed a significant preference for TA+β-infected plants to uninfected Nb plants under white light (Fig 1B). Interestingly, the whiteflies did not show preference for TA+β-infected plants under darkness (Fig 1B). We previously demonstrated that βC1 protein encoded by TYLCCNB is involved in host preference of whitefly [22]. More whiteflies were attracted to TA+β-infected plants compared with the βC1 betasatellite mutant virus (TA+mβ)-infected plants under white light, but there were no significant changes of whitefly preference between TA+β-infected plants and TA+mβ-infected plants under darkness (Fig 1B), suggesting that the viral βC1-mediated whitefly preference is light-dependent.

We next performed whitefly two-choice assays using *βC1* transgenic Nb plants (βC1/Nb) in various light conditions. Due to the complicity of the tripartite interactions, we applied mono-chromatic red-light to represent high red: far-red (R:FR) light ratio and also far-red light to represent low R: FR, the latter mimics the poor light condition of plant competition for light. Different monochromatic light sources were used to determine which wavelength of light is essential for whitefly attraction. Whiteflies were more attracted to the βC1/Nb plants compared to wild-type Nb plants only under red light and white light, not in darkness, far-red light and blue light (Figs 1C and S1A). Moreover, red light-induced whitefly attraction from βC1/Nb plants was disrupted by far-red light. These whitefly preference results under monochromatic lights agree with the field experiments [31], which encouraged us to design other experiments for explicit mechanisms of light effect on the tripartite interactions. To further confirm these results with another host plant, transgenic *Arabidopsis* plants expressing *βC1* (βC1/At) were used to perform whitefly two-choice assays and the same result as in βC1/Nb plants was observed (Fig 1D). Moreover, wild-type Col-0 plants under red light conferred stronger repellence to whitefly than that under darkness on wild-type Col-0 plants (S2A Fig). These results demonstrate that red light plays a crucial role in whitefly preference for *βC1*-expressing plants.

Hemipterans lack red light photoreceptors, so red light likely cannot directly affect whitefly behaviors [32,33]. We thus hypothesized that signals from the host plant mediate the red light-induced changes in whitefly preference for plants expressing *βC1*. Our previous work showed that TYLCCNB βC1 contributes to the suppression of JA-regulated terpene biosynthesis and renders virus-infected plant more attractive to its whitefly vector [22]. Therefore, we examined the expression levels of *TPS* genes under various monochromatic light conditions in

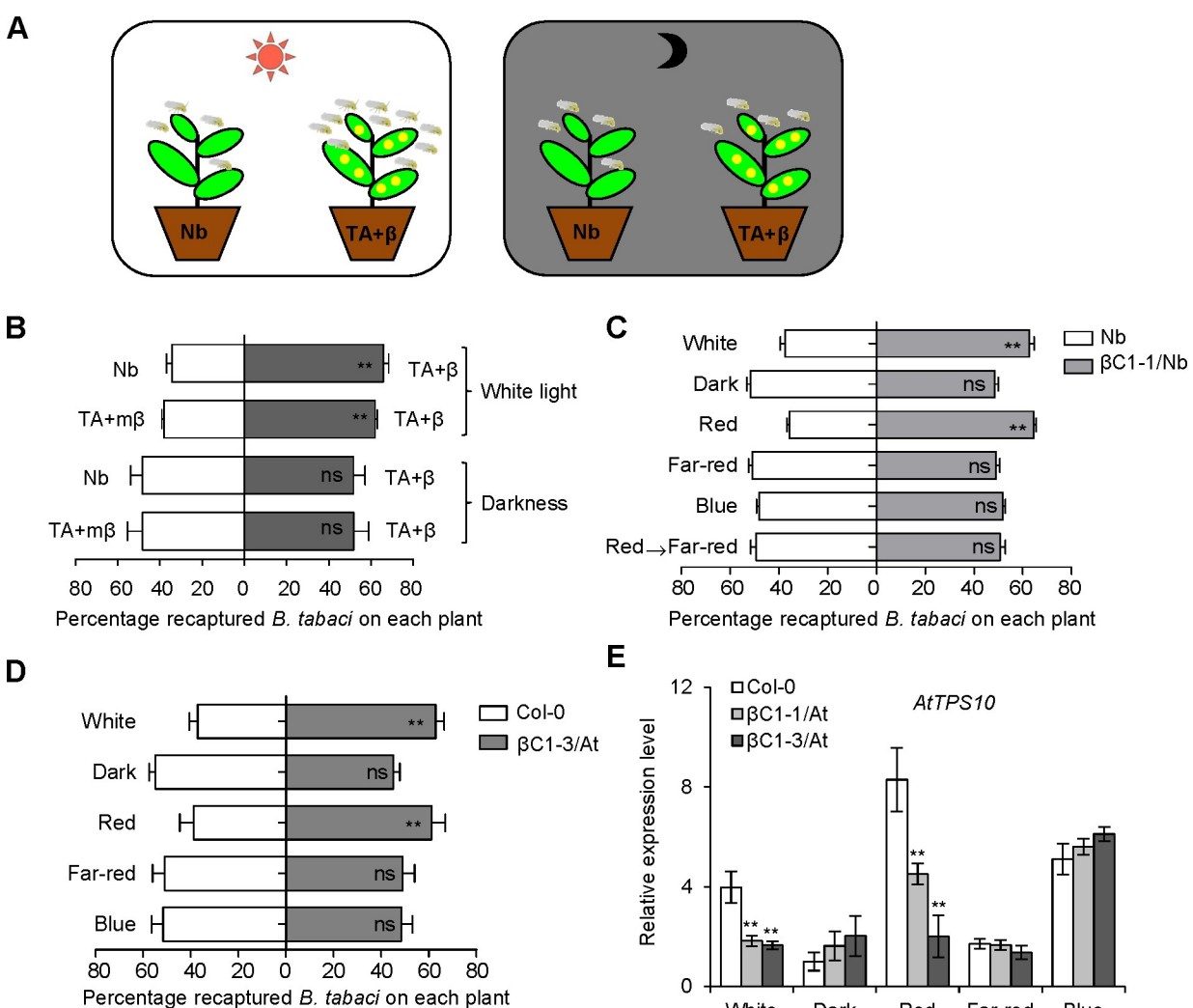

**Fig 1. Begomoviral βC1-mediated whitefly preference is light-dependent. (A)** The schematic diagram of whitefly preference on plant leaves. Control plants (Nb) or virus-infected plants (TA+β) were exposed to white light (left panel) or dark condition at 25°C (right panel) for whitefly preference experiment. **(B)** Whitefly preference (as percentage recaptured whiteflies out of 200 released) on uninfected *N. benthamiana* (Nb, mock) and plants infected by TA+β, or on plants infected by TA+β and a mutant βC1 (TA+mβ) in the white light or under darkness. Values are mean + SD (n = 6). **(C-D)** Whitefly preference on wild-type Nb and βC1 transgenic Nb plants (βC1-1/Nb) (C) or wild-type Col-0 and βC1 transgenic *Arabidopsis* plants (βC1-3/At) (D) in response to white, dark, red, far-red, and blue light. Plants were placed under darkness for 24 h, followed by a 2 h light exposure and then performed whitefly choice experiments. Red→Far-red indicates that plants were firstly kept in darkness for 24 h, followed by a 2 h red light exposure, and then transferred to far-red light for 2 h. Values are mean + SD (n = 6). In B-D, asterisks indicate significant differences between different treatments or lines (**, P< 0.01; ns, no significant differences; the Wilcoxon matched pairs test). **(E)** Relative expression levels of *AtTPS10* in Col-0 and two βC1/At plants (βC1-1/At and βC1-3/At) under different light conditions. Values are mean ± SD (n = 3) (*, P< 0.05; **, P< 0.01; Student's *t*-test). The light was supplied by LED light sources, with irradiance fluency rates of: white (80 μmol m$^{-2}$ sec$^{-1}$), blue (15 μmol m$^{-2}$ sec$^{-1}$), red (20 μmol m$^{-2}$ sec$^{-1}$), and far-red (2 μmol m$^{-2}$ sec$^{-1}$).

*Arabidopsis*. Only red light could induce the βC1-mediated suppression of *AtTPS10*, *AtTPS14*, and *AtTPS21* expression (Figs 1E, S1B, and S1C). We also found that red light induced higher expression of *AtTPS10* than darkness (S2B Fig). These results reveal that βC1 inhibits the transcription of *TPS* genes in a red light-dependent manner.

To explore the potential mechanism underlying the interaction between βC1 and plant signaling under various light conditions, we first excluded the possible roles of light on the

subcellular localization of βC1 protein or its transcript levels (S3A and S3B Fig). We found that the fluorescences from YFP-fused βC1 in white light were more than that in darkness, suggesting that light promotes the accumulation of βC1 (S3A Fig). We further observed that the abundance of βC1 not YFP protein was higher under red light than under darkness, far-red light and blue light (Figs 2A and S3C). Furthermore, we detected the accumulation of βC1 protein in two stable transgenic *Arabidopsis* lines expressing *βC1* (*35S:myc-βC1* #1 and #2). The results show that compared to darkness, white light or red light promotes the stability of βC1

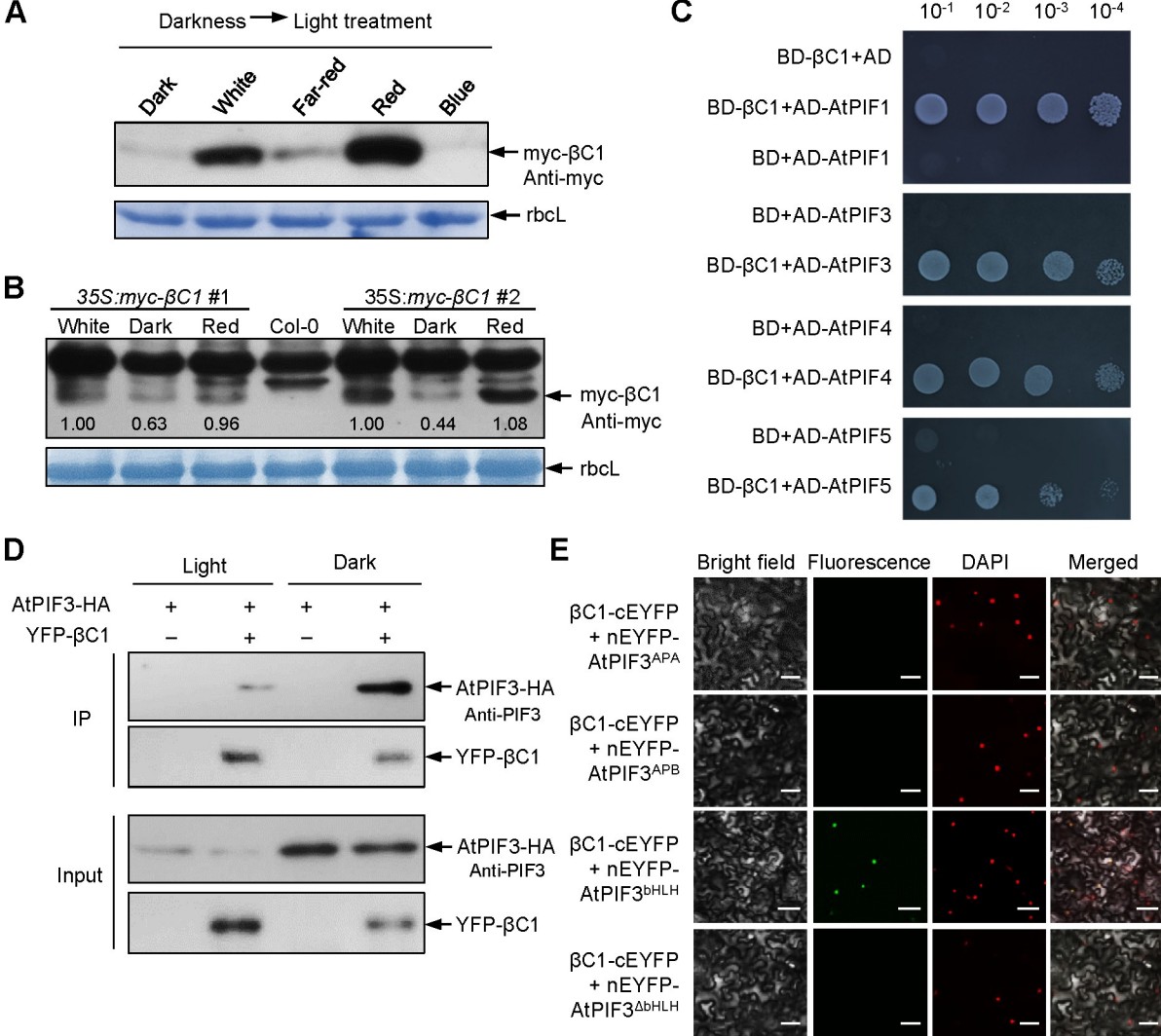

**Fig 2. Light-dependent stability of βC1 interacts with the bHLH domain of phytochrome-interacting factors (PIFs). (A)** Accumulation of βC1 proteins in Nb plants after different light treatments for 2h. Plants were agroinfiltrated with *35S:myc-βC1*, incubated in the dark for 60 h, and followed by a 2 h light exposure. Samples were detected by immunoblot analysis using anti-myc antibody. Stained membrane bands of the large subunit of Rubisco (rbcL) were used as a loading control. **(B)** Accumulation of βC1 proteins in two stable transgenic lines (*35S:myc-βC1* #1 and #2). Plants were placed under darkness for 24 h, followed by a 2 h light exposure. **(C)** Interaction between βC1 and *Arabidopsis* PIFs (AtPIF1, AtPIF3, AtPIF4 or AtPIF5) in the yeast two-hybrid system. The empty vectors pGAD424 and pGBT9 were used as negative controls. **(D)** Co-IP analysis of AtPIF3-HA and YFP-βC1 interaction in the normal light and in darkness. AtPIF3-HA was detected by a western-blot with anti-PIF3 antibody. **(E)** BiFC analysis of AtPIF3 derivative interaction with βC1 protein. The EYFP fluorescences were only observed owing to complementation of βC1-cEYFP with nEYFP-AtPIF3[bHLH] in the normal light. ΔbHLH indicates deletion of bHLH domain in AtPIF3. Scale bars = 50 μm.

protein in stable transgenic plants (Fig 2B). The profile of protein accumulation offered an explanation for the βC1-induced host preference in a red-light dependent manner.

## βC1 interacts with PIFs

To explore how light signal influences βC1-induced whitefly attraction, we used a yeast two-hybrid system to screen for βC1 interactors in an *Arabidopsis* cDNA library. This identified AtPIF3, which was first identified to act in the light transduction pathway and later as multiple signaling integrator [34], as a new βC1-targeted host factor. We next confirmed that βC1 interacts with all four of the *Arabidopsis* PIF-quartet proteins (AtPIF1, AtPIF3, AtPIF4, or AtPIF5) by yeast two-hybrid and bimolecular fluorescence complementation (BiFC) assays (Figs 2C and S4A), further co-immunoprecipitation (CoIP) assay confirmed the interaction between βC1 and AtPIF3 *in vivo* under both dark and light conditions (Figs 2D and S4B). Taken together, these data suggest that βC1 interacts with PIFs independent of light in plants.

PIFs contain a conserved bHLH domain that binds to DNA and mediates dimerization with other bHLH transcription factors to regulate downstream signaling, and another Active Phytochrome A/B-binding domain that interacts with phyA and phyB to sense upstream signaling [19]. To localize βC1-interacting domains, we constructed different AtPIF3 deletion derivatives fused with nEYFP and performed BiFC assays with βC1-cEYFP (S4C Fig). The AtPIF3 fragment that contained the bHLH domain was sufficient to interact with βC1 *in vivo* (Fig 2E), indicating that the interaction between βC1 and PIFs may influence the downstream signaling integrator roles of PIFs in cells.

## The PIFs mediate defense against whitefly in *Arabidopsis*

To investigate whether PIFs are involved in plant defense against insect vectors, we performed whitefly bioassays using Col-0 and *AtPIF3*-overexpressing (*AtPIF3-OE*) transgenic plants. Whiteflies laid fewer eggs and exhibited slower pupa development on *AtPIF3-OE* transgenic plants than that on Col-0 plants (Fig 3A and 3B). Conversely, they laid more eggs and exhibited faster pupa development on *pifq* (*pif1/3/4/5*) quadruple mutant than that on Col-0 plants (Fig 3C and 3D), an observation is similar to that with *βC1*-expressing *Arabidopsis* plants [22]. These data suggest that PIFs are involved in plant defense against whitefly vector.

Since TYLCCNB βC1 involved in whitefly preference in light-dependent (Fig 1) and interacts with plant PIFs (Fig 2), we performed whitefly two-choice assays to examine whether the PIFs have the same effects on whitefly preference as βC1. Consistent with the results with *βC1*-expressing *Arabidopsis* plants, the *pifq* quadruple mutants were higher attractive to whiteflies than Col-0 plants under white or red light (Fig 3E). The transcriptional levels of *TPS* genes (*AtTPS10*, *AtTPS14* and *AtTPS21*) were significantly repressed in the *pifq* mutant compared to those in Col-0 plants under white or red light (Figs 3F, S5A and S5B). Moreover, compared with YFP@βC1-1/Nb plants (normalized to 100%), overexpression of *AtPIF3* or *AtPIF4* approximately decreased whitefly attraction rate to 50% under white light (Fig 3G and 3H), suggesting the overexpression of *PIFs* could partially rescue the susceptibility of *βC1* overexpression plants against whitefly. Taken together, these results imply that PIFs control the plant terpene-based host preference to whitefly in a light-dependent manner.

## βC1 suppresses PIFs activity by interfering with its dimerization

PIFs are bHLH transcription factors that directly regulate gene expression by binding to a core G-box motif (CACGTG) and G-box-like motif (CANNTG) [34,35]. We wondered whether PIFs directly regulate the expression of *TPS* genes and involve in the terpene-mediated whitefly defense response. There are some G-box-like elements (CANNTG) in the promoter of *TPS*

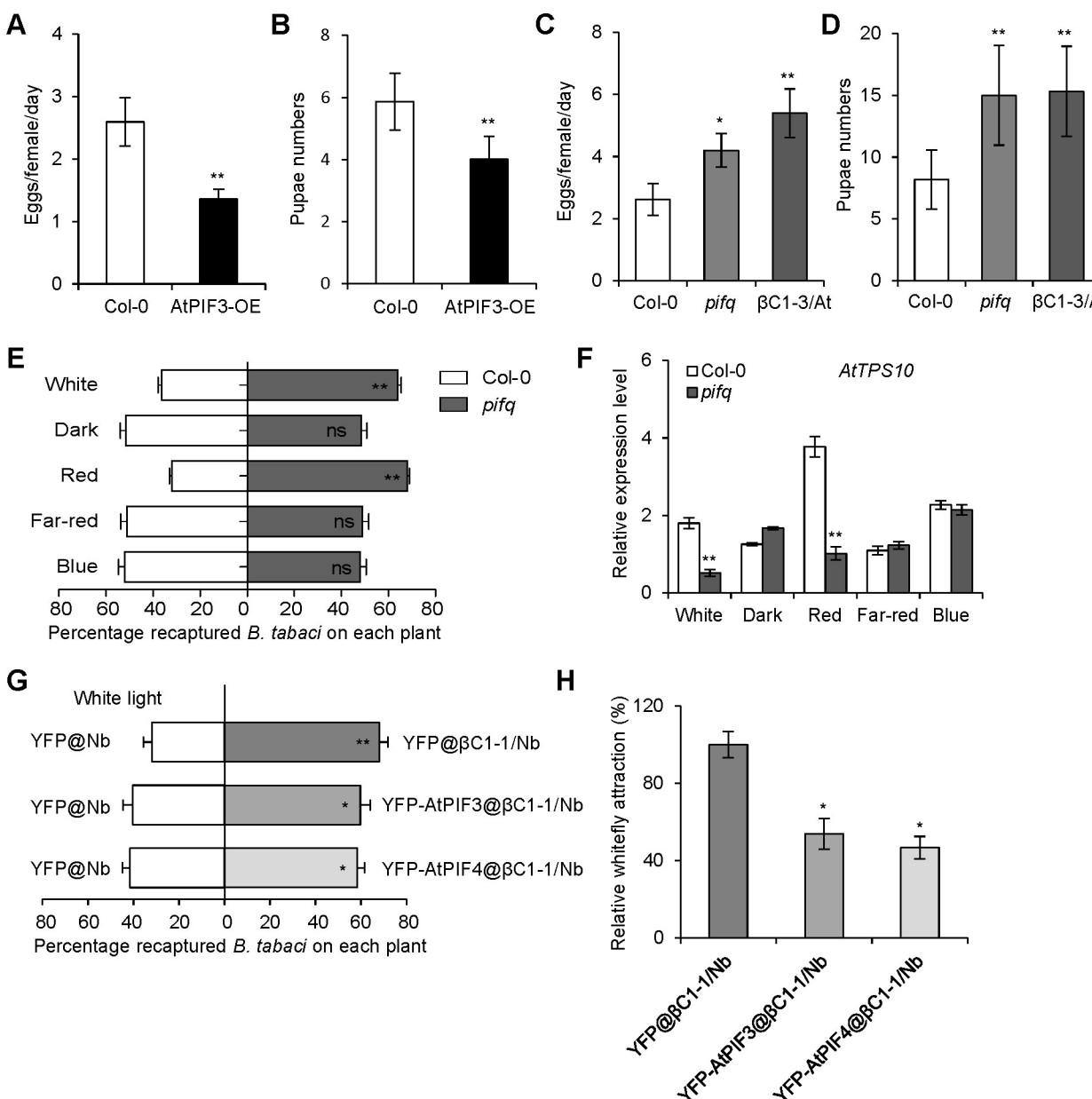

**Fig 3. *Arabidopsis* PIFs confer tolerance to whitefly vector.** (A) Number of eggs laid per female whitefly per day on Col-0 and *AtPIF3*-overexpressing (AtPIF3-OE) transgenic plants. (B) Pupae numbers of whiteflies on Col-0 and AtPIF3-OE transgenic plants. (C) Number of eggs laid per female whitefly per day on Col-0, *pifq* or βC1-3/At plants. (D) Pupae numbers of whiteflies on Col-0, *pifq* or βC1-3/At plants. In A-D, values are mean ± SD (n = 8). Asterisks indicate significant differences of whitefly performance between Col-0 and mutant plants (*, $P < 0.05$; **, $P < 0.01$; Student's *t*-test). (E) Whitefly preference on Col-0 and *pifq* mutant plants in response to white, dark, red, far-red, and blue light. The plants were placed in darkness for 24 h prior to the 2 h different light treatments. Values are mean + SD (n = 6) (**, $P < 0.01$; ns, no significant differences; the Wilcoxon matched pairs test). (F) Relative expression levels of *AtTPS10* in Col-0 and *pifq* mutant plants after a 2 h treatment of different lights. Values are mean ± SD (n = 3) (**, $P < 0.01$; Student's *t*-test). (G) Whitefly preference on transiently expressing *AtPIFs* in βC1-1/Nb and Nb plants with *YFP* overexpression under white light. Values are mean + SD (n = 6) (*, $P < 0.05$; **, $P < 0.01$; the Wilcoxon matched pairs test). (H) Relative whitefly attraction in transiently expressing *AtPIFs* in βC1-1/Nb plants. Relative whitefly attraction in YFP@βC1-1/Nb plants and YFP-AtPIFs@βC1-1/Nb was compared with those in YFP@/Nb plants. Whitefly attraction of YFP@βC1-1/Nb compared to YFP@/Nb plants was set as 100%. Values are mean ± SD (n = 6) (*, $P < 0.05$; Student's *t*-test).

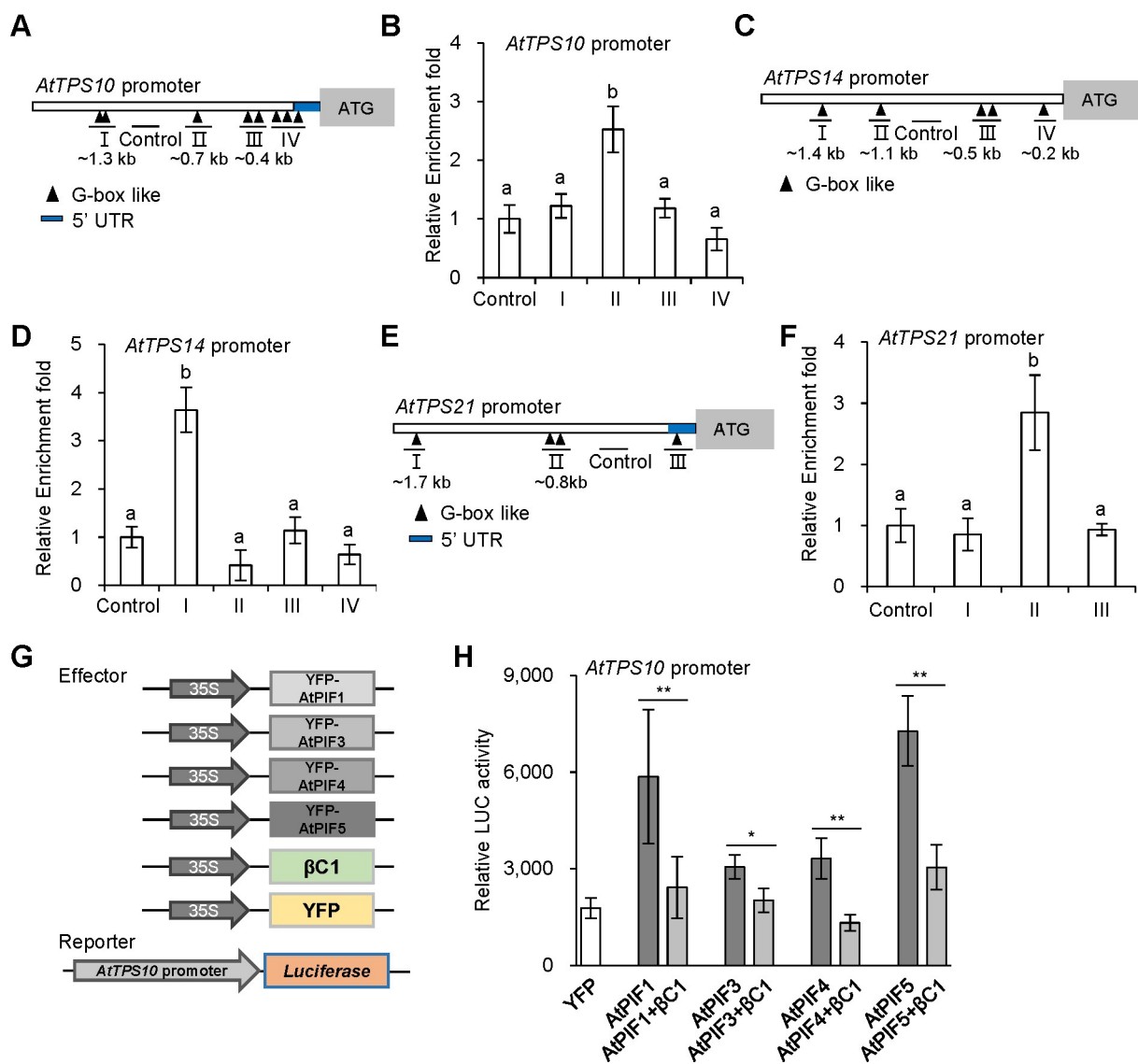

**Fig 4. *Arabidopsis* PIFs directly regulate *TPS* genes and βC1 suppresses transcriptional activity of PIFs. (A, C** and **E)** Schematic diagrams of *AtTPS10* promoter (A), *AtTPS14* promoter (C) and *AtTPS21* promoter (E). The black triangles represent G-box like motifs. A fragment of the four lines (I, II, III and IV), as indicated by the triangles was amplified in ChIP assay. The fragment without G-box like motifs was used as a control. The end positions of each fragment (kb) relative to the transcription start site are indicated below. UTR, untranslated region. **(B, D** and **F)** Fold enrichment of YFP-AtPIF3 associated with each of the DNA fragments of *AtTPS10* promoter (B), *AtTPS14* promoter (D) and *AtTPS21* promoter (F) in ChIP assay. Values are mean ± SD (n = 4). The same letters above the bars indicate lack of significant difference at the 0.05 level by Duncan's multiple range test. **(G)** Schematic diagram showed effector and reporter constructs used in H. *AtTPS10* promoter: *luciferase* (*LUC*) was used as a reporter construct. *35S promoter*-driven *YFP*, *AtPIFs*, and *βC1* were used as effector constructs. **(H)** Effects of βC1 on transcriptional activity of each AtPIFs (AtPIF1, AtPIF3, AtPIF4, or AtPIF5) on *AtTPS10* promoter under white light. Values are mean ± SD (n = 8) (\*, P< 0.05; \*\*, P< 0.01; Student's *t*-test).

genes, which are distributed in several regions (Fig 4A, 4C and 4E). We performed a chromatin immunoprecipitation (ChIP) assay using *AtPIF3-OE* plants. Quantitative PCR analysis showed that region II (one G-box-like motif 0.7 kb upstream of the transcription start site) of *AtTPS10*, region I of *AtTPS14* (one G-box-like motif 1.4 kb upstream of the transcription start site) or region II of *AtTPS21* (two G-box-like motif 0.8 kb upstream of the transcription start site) was significantly enriched in *AtPIF3-OE* lines relative to Col-0 plants (Fig 4B, 4D and 4F).

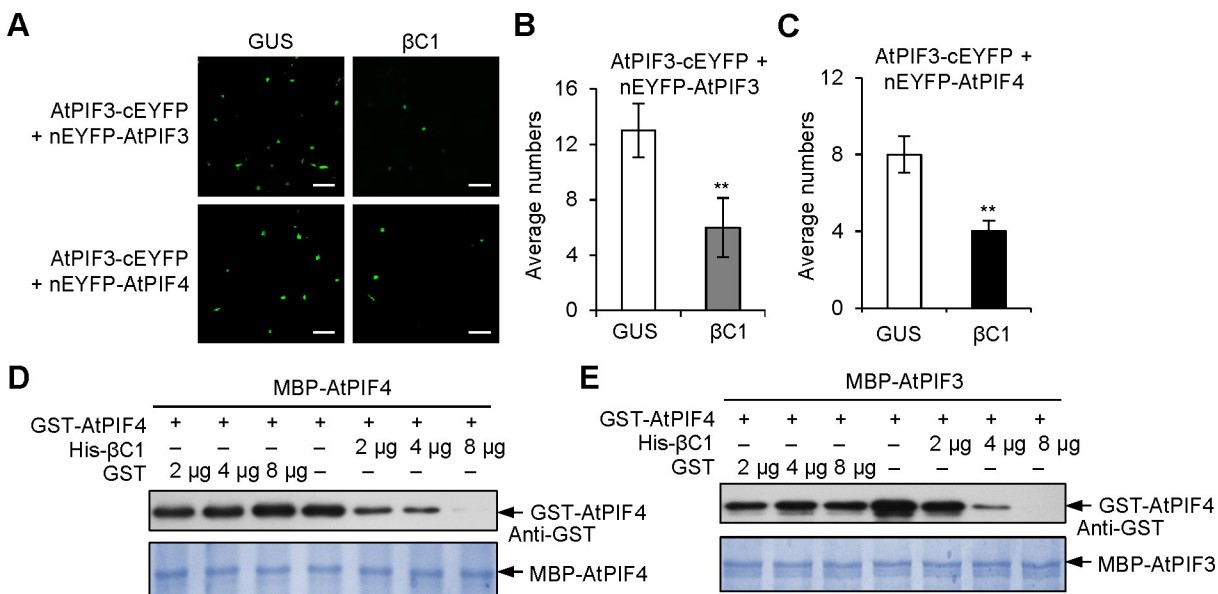

**Fig 5. βC1 interferes with the dimerization of PIFs. (A)** Modified BiFC competition assays. The EYFP fluorescence was detected after co-expression of GUS + AtPIF3-cEYFP + nEYFP-AtPIF3 (GUS), βC1 + AtPIF3-cEYFP + nEYFP-AtPIF3 (βC1), or GUS + AtPIF3-cEYFP + nEYFP-AtPIF4, βC1 + AtPIF3-cEYFP + nEYFP-AtPIF4. Scale bars = 50 μm. **(B-C)** Average numbers of EYFP fluorescence show effects of βC1 on the formation of AtPIF3-AtPIF3 homodimers (B) and AtPIF3-AtPIF4 heterodimers (C). Values are mean ± SD (n = 8) (\*\*, P< 0.01; Student's *t*-test). **(D-E)** GST pull-down protein competition assays. The indicated protein amount of His-βC1 or GST was mixed with 2 μg of GST-AtPIF4 and pulled down by 2 μg of MBP-AtPIF4 (D) or 2 μg of MBP-AtPIF3 (E). Immunoblots were performed using anti-GST antibody to detect the associated proteins. Membranes were stained with Coomassie brilliant blue to monitor input protein amount.

These data indicate that AtPIF3 directly binds to the promoter of *TPS* genes and regulates its expression in *Arabidopsis*.

Next, we examined whether βC1 affects the trans-activity of PIFs via a construct containing the *AtTPS10* promoter with *luciferase* (*LUC*) as a reporter, and YFP-AtPIFs (AtPIF1, AtPIF3, AtPIF4, or AtPIF5) as effectors (Fig 4G). *AtTPS10* promoter: *LUC* was transiently expressed with the indicated effector plus βC1 in Nb leaf cells. Fig 4H shows that each of AtPIFs (AtPIF1, AtPIF3, AtPIF4, and AtPIF5) significantly increased the LUC activity, whereas βC1 decreased AtPIFs-induced LUC activity at different degrees. Therefore, the interaction between βC1 and PIFs blocked the ability of PIFs to regulate *TPS* genes.

PIF3 activates downstream gene expression by forming homodimers and heterodimers with other PIF-related bHLH transcription factors [19]. The interaction between βC1 and the bHLH domain of AtPIF3 (Fig 2E) raised the possibility that βC1 competes with the bHLH domain to interfere with AtPIFs dimerization. A modified BiFC assay was used to test this hypothesis. In cells co-expressing βC1, the interaction signal strength of AtPIF3-AtPIF3 or AtPIF3-AtPIF4 decreased to approximately half of its original intensity (Figs 5A–5C and S5C), suggesting that βC1 may interfere with PIF dimerization. Moreover, *in vitro* competitive pull-down assays showed that βC1 interferes with homodimerization of AtPIF4-AtPIF4 and heterodimerization of AtPIF3-AtPIF4 (Fig 5D and 5E). These results provide evidences that βC1 blocks PIFs dimerization by competing with the binding bHLH domain and thereby suppresses the trans-activity of PIFs.

### PIFs interact with MYC2 to integrate light and JA signals and coordinately regulate host preference of whitefly

PIF-quartet integrates signals from multiple signaling pathways, including light and JA signals, to respond to the diverse stresses and developmental processes [19,34,36]. Previous study has

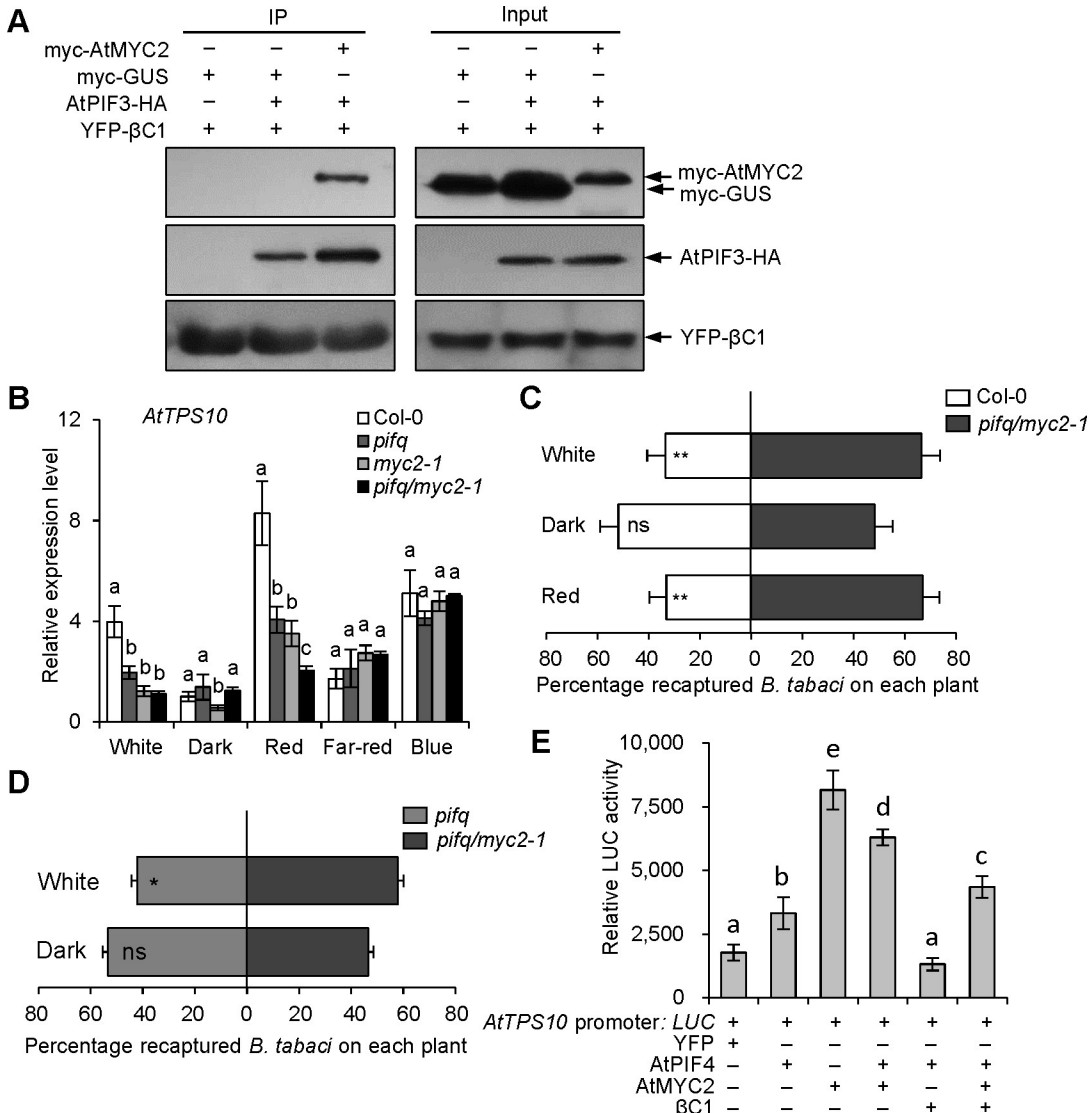

**Fig 6.** *Arabidopsis* **PIFs and MYC2 transcription factors synergistically regulate** *AtTPS10* **transcription and whitefly attraction.** (**A**) Co-IP analysis of YFP-βC1, AtPIF3-HA and myc-MYC2 interaction in a complex. The myc-GUS was used as a negative control. (**B**) Relative expression levels of *AtTPS10* in Col-0, *pifq*, *myc2-1* and *pifq/myc2-1* mutant plants after a 2 h treatment with different light. Values are mean ± SD (n = 3). (**C-D**) Whitefly preference on *pifq/myc2-1* and Col-0 (C) or *pifq* mutant (D) in response to white, dark and red light. Plants were placed under darkness for 24 h, followed by a 2 h light exposure and then performed whitefly choice experiments. Values are mean + SD (n = 6) (*, P< 0.05; **, P< 0.01; ns, no significant differences; the Wilcoxon matched pairs test). (**E**) Effects of βC1 on trans-activation activity of AtPIF4 or AtMYC2 on *AtTPS10* promoter under white light. *AtTPS10* Promoter: *LUC* was used as a reporter construct. YFP, AtPIF4, AtMYC2 and βC1 were used as effector constructs. Values are mean ± SD (n = 8). In A and B, the same letters above the bar indicate lack of significant differences at the 0.05 level in Duncan's multiple range test.

reported that AtPIF4 interacts with AtMYC2, and JA inhibits the function of PIF4 partially through MYC2 in *Arabidopsis* [37]. To confirm that JA and light signaling work cooperatively to regulate plant defense against whitefly, we firstly investigated whether MYC2 associates with PIFs in plants. BiFC assays showed that AtMYC2 interacts with AtPIF3 and AtPIF4 (S6A Fig). And Co-IP assays confirmed that βC1, AtMYC2 and AtPIF4 proteins are in a complex (Fig 6A). Thus, we generated a *pifq/myc2-1* mutant by crossing the *pifq* mutant with the *myc2-*

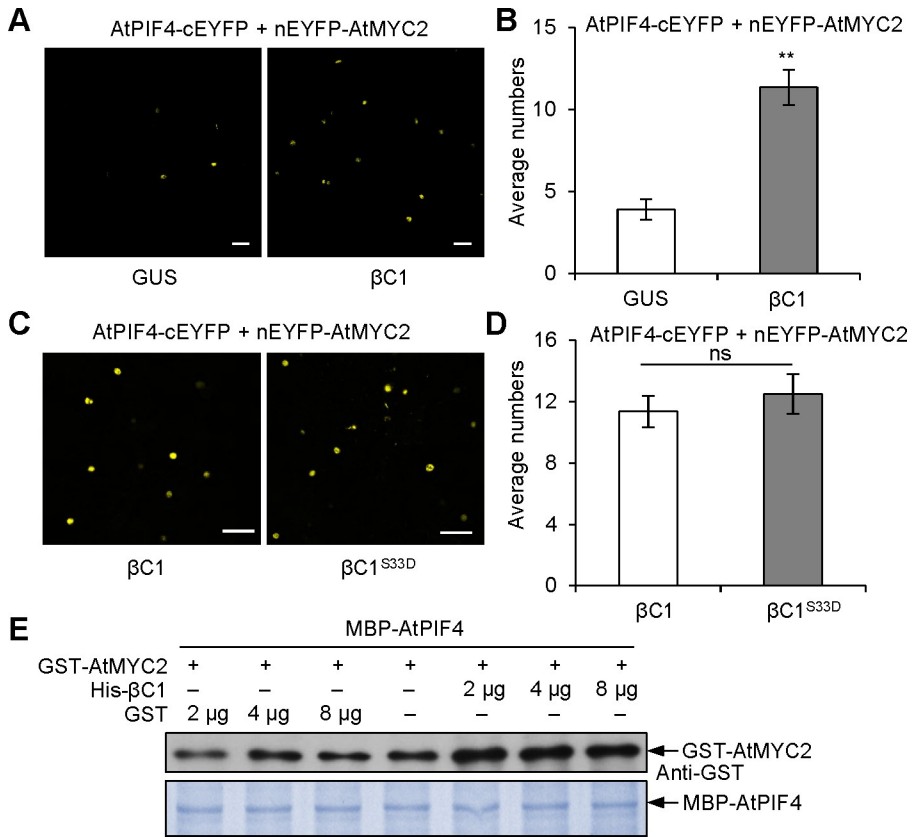

**Fig 7. βC1 promotes the interaction between AtPIF4 and AtMYC2.** (**A**) Modified BiFC competition assays. The EYFP fluorescences were detected using co-expression of AtPIF4-cEYFP + nEYFP-AtMYC2 with or without βC1 under normal light. Scale bars = 50 μm. (**B**) Effects of βC1 on the interaction between AtPIF4 and AtMYC2. Values are mean ± SD (n = 8) (\*\*, P< 0.01; Student's *t*-test). (**C**) Modified BiFC competition assays. The EYFP fluorescences were detected using co-expression of AtPIF4-cEYFP + nEYFP-AtMYC2 with βC1 or a mimic phosphorylation of βC1 (serine-33 to aspartate, βC1$^{S33D}$) under darkness. Scale bars = 50 μm. (**D**) Effects of βC1$^{S33D}$ on the interaction between AtPIF4 and AtMYC2. Values are mean ± SD (n = 8) (ns, no significant differences; Student's *t*-test). (**E**) Protein competition pull-down assay. The indicated protein amount of His-βC1 or GST was mixed with 2 μg of GST-AtMYC2 and pulled down by 2 μg of MBP-AtPIF4. The associated proteins were detected by immunoblots using anti-GST antibody.

*1* mutant. The transcriptional levels of *AtTPS10* in the *pifq/myc2-1* mutant were additively reduced compared to the parental lines under red light conditions (Fig 6B). And the *pifq/myc2-1* mutant was higher attractive to whiteflies than Col-0 and *pifq* mutant under white light (Fig 6C and 6D). The results suggest that AtPIFs and AtMYC2 coordinately regulate the expression of *AtTPS10* and whitefly attraction. Since the individual AtPIF4 or AtMYC2 could directly bind and promote *AtTPS10* expression, we next tested whether the AtPIF4-AtMYC2 interaction has a synergetic effect on downstream genes expression regulation. Unexpected, we found that the heterdimerazation of AtPIF4-AtMYC2 in fact even reduces the transactivation activity when co-expressed with AtPIF4 and AtMYC2 compared to AtMYC2 alone under white light (Fig 6E), indicating an antagonistic effect of heterodimer formation of AtPIF4-AtMYC2 on expression of *AtTPS10*.

Next we tested the effect of viral βC1 on the AtPIF4-AtMYC2 interaction and found that the interaction signal of AtPIF4-AtMYC2 was increased by two-fold when co-expressed with βC1, but not with β-glucuronidase (GUS) (Fig 7A and 7B). To test whether the function of βC1 as a viral suppressor of RNA silencing (VSR) increases the interaction of AtPIF4 and

AtMYC2, we used the mimic phosphorylation of Ser-33 of the βC1 protein (serine to aspartate, βC1$^{S33D}$) as a negative control to perform competitive BiFC experiments. Because it has been reported that βC1$^{S33D}$ protein impairs its functions as VSR [38]. There are no significant differences between βC1 protein and βC1$^{S33D}$ protein in effects on the interaction signal of AtPIF4-cEYFP and nEYFP-AtMYC2 (Figs 7C, 7D sand S6B). These data suggest that βC1 functions as a linker between two bHLH transcription factors AtPIF4 and AtMYC2, which is independent on βC1 known function as VSR. Competitive pull-down assay also supported the idea that βC1 indeed bridges the interaction of AtPIF4-AtMYC2 (Fig 7E). One hypothesis was then raised that the self-interaction of AtPIFs or AtMYCs promotes their transcriptional activity, but the formation of heterodimer of AtPIF4-AtMYC2 inhibits the MYC2 transcriptional activity. Once plants are infected by begomovirus, the linker-βC1 even exacerbates their activities. For that end, we coexpressed βC1 and found that βC1 could dampen the activator activities either by single AtPIF4 or AtMYC2 or coexpression of these two bHLH transcription factors (Fig 6E).

To further explore the function of JA signals in βC1- or PIFs-mediated whitefly host preference, we performed whitefly two-choice assays using *βC1*-expressing and *pifq* mutant plants with MeJA treatment in darkness. The loss of whitefly preference for βC1/At and *pifq* mutant plants under darkness was rescued by MeJA application (Fig 8A and 8B). Accordingly, the expression levels of *AtTPS10*, *AtTPS14* and *AtTPS21* in two βC1/At lines and *pifq* plants were also dramatically decreased by MeJA under darkness (Fig 8C–8H). These results demonstrate that JA and light signals integrally modify begomovirus-whitefly mutualism.

## Discussion

As the Earth warms, we need to be able to predict what conditions will be at risk for infectious diseases because prevention is always superior to reaction. The disease triangle, pathogen-host-environment, is used to understand how disease epidemics can be predicted, restricted or controlled [39]. There is increasing evidence suggesting that environmental factors including light are important mediators of plant defenses during plant-pathogen interactions [14,34]. However, the ability of plant pathogens in using effectors to disturb or co-opt host light signaling to promote infection has not been well explored. Plant defense signals function as players or pawns in plant-virus-vector interactions [40], PIFs are key signal integrators in regulating plant growth and development [16,34]. In recent years, PIFs have been shown to regulate plant defense response [34,41]. There are several evidences to show that PIFs have critical roles in plant developmental and abiotic/biotic defenses not only on dark but also under light conditions. First several interacting partners involved various physiological processes have been reported for PIFs in light condition in previous studies, eg. DELLA, HY5, and freezing tolerance related APETALA2 (AP2) family transcription factor C-repeat binding factor (CBF) [42–44]. Second, a few reports have been shown that PIF4 and PIF5 proteins accumulated to higher levels in light than in dark though PIF3 accumulated to the highest level in dark [45,46]. Here, we provide evidence that PIF proteins also act as direct positive regulators in plant defense against whitefly vector (Fig 4). First, the *pifq* quadruple mutants with lower expression of *TPS* genes are more sensitive to whiteflies than Col-0 plants under white or red light (Figs 3E, 3F, S5A and S5B). Second, the *AtPIF3*-overexpression confers enhanced *Arabidopsis* resistance to whitefly, and PIFs deficiency in *pifq* mutant promotes whitefly performance in *Arabidopsis* (Fig 3A–3D). Third, the *PIFs* overexpression in *βC1*-expressing plants can partially rescue the susceptibility of *βC1* overexpression plants against whitefly under white light (Fig 3G and 3H). Fourth, the additional expression of *βC1* in the background of *pifq* mutant (βC1/*pifq* mutant) could further increase whitefly attraction compared with *pifq* mutant in the light, but not

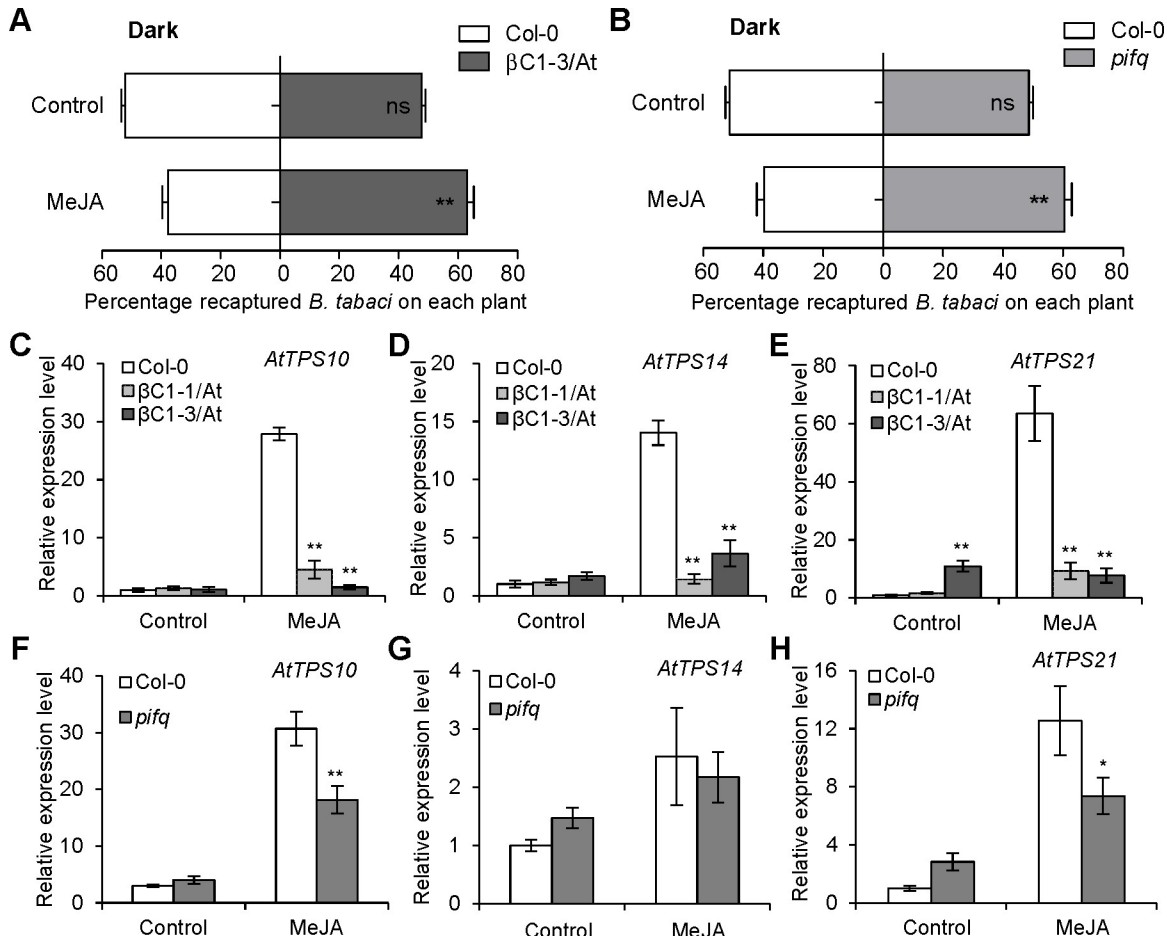

**Fig 8. Light and JA signals synergistically regulate whitefly host preference. (A-B)** Whitefly preference on Col-0 and βC1-3/At plants (A) or Col-0 and *pifq* mutant plants (B) with or without MeJA treatment under darkness. Values are mean ± SD (n = 6) (**, P< 0.01; ns, no significant differences; the Wilcoxon matched pairs test). **(C-E)** Relative expression levels of *AtTPS10* (C), *AtTPS14* (D), and *AtTPS21* (E) in Col-0 and two βC1 transgenic *Arabidopsis* lines with or without MeJA treatment under darkness. Values are mean ± SD (n = 3) (**, P< 0.01; Student's *t*-test). **(F-H)** Relative expression levels of *AtTPS10* (F), *AtTPS14* (G), and *AtTPS21* (H) in Col-0 and *pifq* mutant with or without MeJA treatment under darkness. Values are mean ± SD (n = 3) (*, P< 0.05; **, P< 0.01; Student's *t*-test).

darkness (S7 Fig). These partial rescue data are consistent with the fact that βC1 have multiple targets on JA signal pathways MYC2, AS1 and SKP1 [22,47,48]. Especially, PIFs function together with MYC2 to control plant olfactory defenses against vector insects in a light-dependent mode. Fifth, the co-presence of βC1 and PIFs on vascular tissues where the βC1 and whitefly live with suggests begomoviral βC1 proteins bound to the plant vasculature-specific expression of transcription factor PIF4, similar as another interactor of βC1-WRKY20 in the vascular tissue [10,49]. Therefore begomoviral βC1 protein performs a successful counter-defense by hijacking PIFs proteins. On one hand, βC1 interacts with PIFs and suppresses trans-activity of PIFs by interfering with its dimerization (Figs 2, 4 and 5); on the other hand, βC1 utilizes whitefly vector to decrease the *PIFs* transcription induced by begomovirus in host plants (S8 Fig). Consequently, begomovirus suppresses PIFs-mediated plant defense to enhance vector transmission.

Most of plant arboviruses attract their insect vectors by modulating plant host-insect vector specific recognition. Light modulates communications of plant-insect through a combination of olfactory and visual cues comprehensively [14,50]. Similarly as our current results, red light

seems essential for a terpenoid volatile based-attraction to Huanglongbing host plant for the vector insect Asian Citrus Psyllid, which transmits the casual bacterial pathogen *Candidatus Liberibacter* [51]. MYC2 and its homologs have been characterized as a few known regulators in terpene biosynthesis mainly during day time [21–23], since it stabilizes by light but destabilizes in darkness [52]. A recent study demonstrated that the MYC2 protein in JA signaling pathways interacts with PIF4 [37]. Here, we likewise show that AtPIF3 and AtPIF4 all interact with AtMYC2. Both AtPIF3 and AtMYC2 can directly bind to the *TPS* promoter and regulate its expression (Fig 4) [22, Fig 4A–4E]. Compared to AtMYC2 which binds to the proximal end of the *AtTPS10* promoter, one G-box-like motif 0.4 kb upstream of the transcription start site, binding G-box like motif of AtPIF3 is in the distal end of the *TPSs* promoter, thus the *AtTPS10* transcriptional activity of AtMYC2 is higher than that of PIFs (Fig 6E). *AtTPS10* expression is significantly reduced in *pifq/myc2-1* quintuple mutant compared with parental single *pifq* or *myc2-1* mutant under red light (Fig 6B). The *pifq/myc2-1* quintuple mutant attracts more whiteflies than *pifq* mutant under white light (Fig 6D). Meanwhile transcript accumulation of *TPS* genes (*AtTPS10*, *AtTPS14* and *AtTPS21*) does not show a circadian rhythm in *Arabidopsis* (S9A Fig). The expression of PIFs and MYCs was complemented and balanced regulation (S9B and S9C Fig). These results suggest that PIFs and MYC2 synergistically regulate terpene biosynthesis in two paralleled pathways, and they are balanced to regulate the expression of *AtTPS* genes and whitefly attraction in white light and darkness. Furthermore, the mechanism of PIFs-regulated *TPS* genes expression in dark is complementary to the MYC2-regulated counterpart in light. Since PIFs are much stable in the night, PIFs may control the ecological interactions of plant-insects in night by regulating the chemical communication, esp. for these night blooming plants and behaviors of nighttime feeding insects [53]. In addition, PIF-like genes are highly conserved and they have been existed before the water-to-land transition of plants [16,54]. It will be of interest to examine possible defensive roles in PIF homologs in other plants. Our findings indicate prospects for biotechnological improvement of crops to improve yield and immunity simultaneously through editing and regulation of *PIFs* genes.

Under red light, PIFs levels/activities are expected to be low in wild-type plants, because PIFs are inactivated by phyB. phyB is the predominant photoreceptor regulating photomorphogenic responses to red light, while phyA is the primary photoreceptor responsible for perceiving far-red light [55,56]. Our results show that the accumulation of βC1 protein was higher in red light than that in far-red light (Fig 2A and 2B). Interestingly, when we treated the Nb plants transiently expressed myc-βC1 protein with continuous red light and far-red light, the βC1 protein was accumulated in red light, but decreased in far-red light (S10 Fig). When the plants in red light again, βC1 protein accumulation has no obvious changes, but reduced again when treated with far-red light (S10 Fig). These results further prove that red light could maintain the stability of βC1 protein, but far-red light promotes the degradation of βC1 protein, which imply that the photoreceptors phyB or phyA might involve in regulation of βC1 stability. This hypothesis needs further research.

Modern anti-arbovirus strategy includes anti-insect netting to disrupt disease transmission. Also more and more countries have adapted greenhouse crop production under protected condition in the past decades. In northern countries this practice often relies heavily on supplemental lighting for year-round yield and product quality. Among the different spectra used in supplemental lighting, red light is often considered the most efficient [2]. It seems like that begomovirus could adapt these serial artificial environmental changes by evolving new role of a known virulence factor to hijcak host internal light signaling. Plant viruses have a small genome in which the encoding proteins especially the virulence factors are frequently multifunctional. βC1 protein is multifunctional and has many host targets for its pathogenesis [29,57], many of which may impact plant-virus and plant-whitefly interactions. It is necessary

to further dissect whether and how other targets of βC1 are also involved in this light-dependent virus pathogenicity in the future. Meanwhile, the data collected here and conclusion we made is based on well-controlled monochromatic light conditions. When extrapolating to natural and agricultural field conditions, it should seriously take into account the real light quality within dense stands in the begomovirus-whitefly-plant tripartite interactions. Nevertheless our data here is significant for understanding of the tripartite interactions and also for arbovirus disease controlling, esp. begomoviral βC1 is adapted to red-light condition, which represents a good light quality, to suppress phytohormone-regulated terpene biosynthesis to attract whitefly insect.

The results in this study can be best summarized by the working model presented in S11 Fig. In this model, homodimerized PIFs or MYC2 binds to the promoter regions of *TPS* genes, resulting in increased *TPSs* transcript levels and terpene biosynthesis. Thus red-light signal and JA signal fine-tune transcription of *TPS* genes to contribute to resistance to whiteflies in uninfected plants (S11A Fig). In begomovirus-infected plants, βC1 inhibits transcriptional activity of PIFs and MYC2 by interfering with their homodimerization and promoting AtPIFs-AtMYC2 heterodimerization. Finally, the decreased terpene synthesis and in turn enhanced whitefly performance increase the probability of pathogen transmission (S11B Fig).

## Materials and methods

### Plant materials and growth conditions

Wild-type or transgenic *Nicotiana benthamiana* plants carrying *35S:βC1* have been reported previously [22,48]. *N. benthamiana* plants grew in an insect-free growth chamber at 25°C with 12 h light/12 h darkness cycle. *Arabidopsis thaliana* wild-type Col-0, *pifq* (*pif1/3/4/5*) [58], *myc2-1* mutant [59], and βC1/At [22] were used in the study. Quintuple *pifq*/*myc2-1* mutant was generated by crossing the corresponding parental single *myc2-1* and quadruple *pifq* homozygous lines. The construct expressing *35S:YFP-AtPIF3* was transformed into Col-0 plants, and generated *AtPIF3*-overexpressing lines (*AtPIF3-OE*). Sterilized seeds were incubated on Murashige and Skoog medium at 4°C for 3 d before being transferred to a growth chamber (22°C with 10 h of light/14 h of darkness cycle).

### Plant treatments

For whitefly two-choice assays and *Arabidopsis TPSs* expression analysis, plants were placed in darkness for 24 h, followed by a 2-h light exposure for two-choice assays. White light, blue light, red light, and far-red light were supplied by LED light sources, the irradiance fluency rates was, white light (80 μmol m$^{-2}$ sec$^{-1}$), blue light (15 μmol m$^{-2}$ sec$^{-1}$), red light (20 μmol m$^{-2}$ sec$^{-1}$), and far-red light (2 μmol m$^{-2}$ sec$^{-1}$). Light intensity was measured with an OHSP-350C illumination spectrum analyzer.

For phytohormones treatments, methyl jasmonate (MeJA) was used to mimic whitefly infestation in *N. bentheamina* and *Arabidopsis* [22]. Three week-old *Arabidopsis* were sprayed with 100 μM MeJA containing 0.01% (v/v) Tween 20. Plant samples were collected at 6 h following treatment. Control plants were treated with 0.01% (v/v) Tween 20 in parallel for the same time period.

### Virus inoculation

*N. benthamiana* plants with four to six true leaves were infiltrated with *Agrobacterium tumefaciens* carrying TYLCCNV and betasatellite DNAβ (isolate Y10) as described previously

[22,60]. Infiltration with buffer or TYLCCNV plus a mutant betasatellite DNA with a βC1 mutation (TA+mβ) was used as a control [60].

## Whitefly bioassays

Whiteflies were collected in the field in Chaoyang District, Beijing, China and were identified as *Bemisia tabaci* MEAM1, B biotype (mtCOI, GenBank accession number MF579701). The whitefly population was maintained in a growth chamber (25˚C, 65% RH) on cotton with a 12 h-light/12 h-dark light cycle.

The whitefly two-choice experiments were performed as described previously [22]. Two plants of selected genotypes with similar size and leaf numbers were firstly kept in darkness for 24 h, and then exposed to specific light for 2 h, and finally placed in an insect cage (30*30*30 cm) with the same light condition. Two hundred adult whiteflies were captured, and then released from the middle of the two plants. After 20 min, the whiteflies settled on each plant were recaptured and the number on each plant was recorded. Six biological replicates were conducted in this experiment.

For whitefly oviposition experiment, three female and three male whitefly adults were released to a single leaf encircled by a leaf cage (diameter, 45 mm; height, 30 mm). All the eggs on the *Arabidopsis* leaves were counted with a microscope after 10 d, and the number of eggs deposited per female was determined. Eight biological replicates were conducted in this experiment.

For the whitefly development experiment, 16 female adults were inoculated to a single leaf encircled by a leaf cage. After 2 d of oviposition, all adults were removed, and the eggs were allowed to develop. All pupae on the *Arabidopsis* leaves were counted with a microscope after 22 d, and the number of pupae per female was determined. Eight biological replicates were conducted in this experiment.

## Yeast two-hybrid analysis

The *Arabidopsis* Mate and Plate Library were used (Clontech, 630487). Full-length protein for βC1 was cloned into the pGBT9 vector to generate BD-βC1 construct. This was then used to screen against the full yeast library via the yeast mating system following the manufacturer's protocol (Matchmaker Gold Yeast Two-Hybrid System, Clontech). To further confirm the interaction between βC1 and AtPIFs, full-length of *Arabidopsis* PIFs was cloned into the pGAD424 vector through LR reaction to generate AD-AtPIFs. The yeast strain Y2HGold was co-transformed with BD-βC1 and AD-AtPIF1/PIF3/PIF4/PIF5 constructs and plated on SD-Leu-Trp selective dropout medium. Colonies were transferred onto SD-Leu-Trp-His plates to verify positive clones. The empty vectors pGBT9 and pGAD424 were used as negative controls.

## Bimolecular fluorescence complementation (BiFC)

Fluorescence was observed owing to complementation of the βC1-fused with the C-terminal part of EYFP and one of AtPIFs-fused with the N-terminal part of EYFP, or complementation of the AtPIF4-fused with the C-terminal part of EYFP and MYC2-fused with the N-terminal part of EYFP. Unfused nEYFP was used as a negative control. Leaves of 3-week-old *N. benthamiana* plants were infiltrated with *Agrobacterial* cells containing the constructs designed for this experiment. Two days after infiltration, fluorescence and DAPI staining were observed by confocal microscopy. Three independent plants were tested in one experiment. The experiment was repeated twice with similar results.

## Co-immunoprecipitation (Co-IP) assay

*A. tumefaciens* strains containing expression vectors of *35S:YFP* and *35S:AtPIF3-HA*, *35S:YFP-βC1* and *35S:AtPIF3-HA*, or *35S:YFP-βC1* and *35S:AtMYC2-HA* were co-injected into 3-week-old *N. benthamiana* leaf cells. YFP was used as negative control, and AtMYC2-HA was used as positive control. After infiltration, plants were maintained in the dark (in order to stabilize PIFs) or normal light conditions for 2 d before protein extraction [61]. Total proteins were extracted from infiltrated leaf patches in 1 ml lysis buffer [50 mM Tris-HCl pH7.4, 150 mM NaCl, 2 mM MgCl$_2$, 10% glycerol, 0.5% NP-40, 1 mM DTT, protease inhibitor cocktail (Roche, 32147600)]. Fifty milligram protein extracts were taken as input, and then the rest extracts were incubated with the GFP-Trap beads (ChromoTek, gta-20) for 1.5 h at 4°C. Immunoblotting was performed with anti-HA and anti-GFP antibodies (TransGen Biotech, HT801-02).

## Pull-down protein competitive interaction assay

The GST- and MBP-fusion proteins were separately purified using Glutathione sepharose (GE Healthcare, 17-5132-01) and Amylose resin (New England Biolabs, E8021S) beads as according to the manufacturer's instructions. His-βC1 fusion proteins were purified using Ni-nitrilotriacetate (Ni-NTA) agarose (Qiagen, 30210) according to the manufacturer's instructions. Indicated amounts of GST or His-βC1 were mixed with 2 μg of MBP-fusion proteins and 50 μL of Amylose resin overnight. After two washes with binding buffer (50 mM Tris-HCl, pH 7.5, 100 mM NaCl, 35 mM β-mercaptoethanol and 0.25% Triton X-100), 2 μg of GST-fusion proteins were added and the mixture was incubated for 3 h at 4°C. Beads were washed 6 times with binding buffer. The associated proteins were separated on 8% SDS-polyacrylamide gels and detected by immunoblots using anti-GST antibody (TransGen Biotech, HT601-02).

## Quantitative RT-PCR

Total RNA was isolated using the RNeasy Plant Mini Kit (Qiagen, 74904), and 2000 ng of total RNA for each sample was reverse transcribed using the TransScript One-Step gDNA Removal and cDNA Synthesis SuperMix (TRAN, AT311-03). Three independent biological samples, each from an independent plant, were collected and analyzed. RT-qPCR was performed on the CFX 96 system (Bio-Rad) using Thunderbird SYBR qPCR mix (TOYOBO, QPS-201). The primers used for mRNA detection of target genes by real-time PCR are listed in S1 Table. The *Arabidopsis Actin2* (At3g18780) mRNA was used as internal control.

## ChIP assay

Transgenic *Arabidopsis* plants expressing *35S:YFP-AtPIF3* and wild-type control Col-0 were used for ChIP assays. *Arabidopsis* seedlings were grown on MS medium for 12 days. 2.5 g of seedlings were harvested and fixed in 37 ml 1% formaldehyde solution under a vacuum for 10 min. Glycine was added to a final concentration of 0.125 M, and the sample was vacuum treated for an additional 5 min. After three washes with distilled water, samples were frozen in liquid nitrogen. ChIP experiments were performed as described using anti-GFP agarose beads (GFP track, gta-20) for immunoprecipitation [62]. The resulting DNA samples were purified with the QIA quick PCR purification kit (Qiagen, 28106). DNA fragments were analyzed by quantitative PCR, with the *Arabidopsis ACTIN2* (At3g18780) promoter as a reference. Enrichments were referred to the *35S:YFP-AtPIF3* against wild-type Col-0 seedlings. Primers of ChIP assays are listed in S1 Table. The experiments were repeated with four independent biological samples, each from independent plants.

## Luciferase activity assay

*AtTPS10* Promoter: *luciferase* was used as a reporter construct. *35S:YFP*, *35S:AtPIF1*, *35S:AtPIF3*, *35S:AtPIF4*, *35S:AtPIF5*, *35S:AtMYC2* and *35S:βC1* were used as effector constructs. Nb leaves were agro-infiltrated with the constructs indicated in each figures. Two days after infiltration, leaves were harvested and the luciferase (LUC) activity of infiltrated leaf cells was quantified by microplate reader as described [22]. Each treatment was repeated eight times in one experiment.

## Protein extraction and western blot

For βC1 stability assays, construct containing *35S:myc-βC1* was infiltrated with *A. tumefaciens* strains (EHA105) and transiently expressed in leaves of four-week-old *N. benthamiana*. Plant samples were placed under different light conditions as indicated as in Fig 2. Total proteins were extracted from infiltrated leaf patches in 1 ml 2×NuPAGE LDS sample buffer (Invitrogen, NP0008) containing 0.05mL/mL β-mercaptoethanol, and protease inhibitor cocktail. Ten milligram protein extracts were taken for immunoblotting with anti-myc antibody (TransGen Biotech, HT101-01).

## Data analysis

Differences in whitefly performance, gene expression levels and average numbers of EYFP fluorescence were determined using Student's *t*-tests for comparing two treatments or two lines. Differences in relative enrichment fold of DNA fragments in the promoter and relative LUC activity were determined using One-way ANOVA, followed by Duncan's multiple range test for significant differences among different lines or different treatments. Differences in whitefly two-choice between different lines were analyzed by Wilcoxon matched pairs tests (with two dependent samples). All tests were carried out with GraphPad Prism.

## Supporting information

**S1 Fig. Begomovirus encodes βC1 to modulate light-regulated plant defense. (A)** Whitefly preference on wild-type Nb and βC1 transgenic Nb plants (βC1-2/Nb) in response to white, dark, red, far-red, and blue light. Plants were placed under darkness for 24 h, followed by a 2 h light exposure and then performed whitefly choice experiments. Red→Far-red indicates that plants were firstly kept in darkness for 24 h, followed by a 2 h red light exposure, and then transferred to far-red light for 2 h. Values are mean + SD (n = 6) (**, P< 0.01; ns, no significant differences; the Wilcoxon matched pairs test). **(B-C)** Relative expression levels of *AtTPS14* (B), and *AtTPS21* (C) in Col-0 and two βC1/At plants (βC1-1/At and βC1-3/At) under different light conditions. Values are mean ± SD (n = 3) (*, P< 0.05; **, P< 0.01; Student's *t*-test). The light was supplied by LED light sources, with irradiance fluency rates of: white (80 μmol m$^{-2}$ sec$^{-1}$), blue (15 μmol m$^{-2}$ sec$^{-1}$), red (20 μmol m$^{-2}$ sec$^{-1}$), and far-red (2 μmol m$^{-2}$ sec$^{-1}$). (TIF)

**S2 Fig. Red light plays a crucial role for plant defense against whitefly. (A)** Whitefly preference (as percentage recaptured whiteflies out of 200 released) on wild-type Col-0 in response to darkness or red light. The plants were placed in darkness for 24 h prior to the 2 h dark or 2 h red light (20 μmol m$^{-2}$ sec$^{-1}$) treatments. Values are mean + SD (n = 6) (**, P< 0.01; the Wilcoxon matched pairs test). **(B)** Relative expression levels of *AtTPS10* in Col-0 plants exposed to darkness or red light. Values are mean ± SD (n = 3). Asterisks indicate significant differences of *AtTPS10* expression in Col-0 plants between under darkness and red light (**, P<

0.01; Student's *t*-test).
(TIF)

**S3 Fig. Light has no visible effect on the subcellular localization of βC1 protein or its transcript levels. (A)** Subcellular localization of YFP-βC1 in *N. benthamiana* under darkness or white light condition. After transient inoculation of *35S:YFP-βC1*, plants were placed in the dark or in the white light for 48 h prior to the observation. Scale bars = 50 μm. **(B)** Relative expression levels of *βC1* in Col-0 plants in response to dark or white light. Values are means ± SD (n = 3). 'ns' indicates no significant differences. **(C)** Accumulation of YFP proteins in Nb plants after different light treatments for 2h. Plants were agroinfiltrated with *35S: YFP*, incubated in the dark for 60 h, and followed by a 2 h light exposure. Stained membrane bands of the large subunit of Rubisco (rbcL) were used as a loading control.
(TIF)

**S4 Fig. βC1 interacts with AtPIFs protein. (A)** Co-IP analysis of AtPIF3-HA and YFP-βC1 interaction *in vivo*. YFP was used as a negative control, while AtMYC2-HA was used as a positive control. All of above interaction experiments were performed in normal light condition. **(B)** *In vivo* BiFC analysis of βC1 interaction with *Arabidopsis* PIFs (AtPIF1, AtPIF3, AtPIF4 or AtPIF5). Fluorescence was observed owing to complementation of the βC1-cEYFP fused protein and nEYFP-AtPIFs fused protein. Nuclei of Nb leaf epidermal cells were stained with DAPI. Unfused nEYFP was used as a negative control. Scale bars = 50 μm. **(C)** Domain structure of AtPIFs proteins. Schematic diagrams of the AtPIFs polypeptide show the location of the consensus basic helix-loop-helix (bHLH) domain, which defines this transcription factor family, as well as the Active Phytochrome A-binding (APA) region and the Active Phytochrome B-binding (APB) region.
(TIF)

**S5 Fig. PIFs regulate the transcript levels of *TPS* genes. (A-B)** Relative expression levels of *AtTPS14* (A) and *AtTPS21* (B) in Col-0 and *pifq* mutant plants after a 2 h treatment of different lights. Values are mean ± SD (n = 3) (**, P< 0.01; Student's *t*-test). **(C)** Detection of protein levels in modified BiFC assay of Fig 5A.
(TIF)

**S6 Fig. AtPIF proteins interact with MYC2. (A)** *In vivo* BiFC analysis of AtMYC2 interaction with AtPIFs (AtPIF3 or AtPIF4) in normal light. Scale bars = 50 μm. **(B)** Detection of protein levels in modified BiFC assay of Fig 7C.
(TIF)

**S7 Fig. The additional expression of βC1 in *pifq* mutant further increases whitefly attraction under white light. (A)** The phenotype of the additional expression of βC1 in *pifq* mutant (βC1/p*ifq* mutant). **(B)** Detection of *βC1* expression in the βC1/p*ifq* mutant by semi-quantitative PCR. **(C)** Whitefly preference on *pifq* mutant and βC1/p*ifq* mutant in response to white light and dark. Values are mean + SD (n = 6) (*, P< 0.05; ns, no significant differences; the Wilcoxon matched pairs test).
(TIF)

**S8 Fig. Begomovirus infection triggers PIFs transcription in *Arabidopsis*.** Relative expression levels of *AtPIFs* in *Arabidopsis* plants. *Arabidopsis* Col-0 plants agroinfiltrated with the infectious clones of TA+β complex at 14 dpi, followed by infestation by whiteflies for 6 h. Total plant RNAs were extracted for qRT-PCR analysis. Uninfected Col-0 plants were used as mock. Values are means ± SD (n = 3). Asterisks indicate significant differences of *AtPIF* genes

expression between mock and infected-Col-0 plants (*, P< 0.05; **, P< 0.01; Student's *t*-test).
(TIF)

**S9 Fig. The expression of *PIFs* and *MYCs* is complemented and balanced regulation. (A)**
The expression pattern of *Arabidopsis TPS10/TPS14/TPS21* is constant during night and day
time. Relative expression levels of *AtTPS* genes in Col-0 under 12 h light/12 h darkness. Values
are mean ± SD (n = 3). **(B)** Relative expression levels of *AtMYC* genes in Col-0 and *pifq* mutant
plants under light. **(C)** Relative expression levels of *AtPIF* genes in Col-0 and *myc2-1* mutant
plants under light. Values are mean ± SD (n = 3). In B-C, asterisks indicate significant differ-
ences of genes expression between Col-0 and mutant plants (*, P< 0.05; **, P< 0.01; Student's
*t*-test).
(TIF)

**S10 Fig. βC1 protein accumulation in continuous red light and far-red light.** Accumulation
of βC1 proteins in Nb plants after treated with continuous red light and far-red light. Plants
were placed under darkness for 60 h, then transferred to continuous red light and far-red light
for 2 h respectively. Samples were detected by immunoblot analysis using anti-myc antibody.
Stained membrane bands of the large subunit of Rubisco (rbcL) were used as a loading con-
trol.
(TIF)

**S11 Fig. A working model of red-light regulated begomovirus-whitefly mutualism. (A)** In
uninfected plant, both plant PIFs and MYC2 mediate the transcription of *TPS* genes by respec-
tively binding to different G-box-like elements of the promoter region, and activate *TPSs* tran-
scription. Thus, red-light signal and JA signal fine-tune transcription of *TPS* genes in plants to
defend against whitefly. **(B)** In begomovirus-infected plants, βC1 interacts with PIFs and
MYC2, and inhibits their transcriptional activity by interfering with their homodimerization
and promoting AtPIFs-AtMYC2 heterodimerization. Finally, the decreased terpene synthesis
and in turn enhanced whitefly performance increase the probability of pathogen transmission.
(TIF)

**S1 Table. DNA primers used in this study.**
(XLSX)

## Acknowledgments

We thank Prof. Peter Quail for providing *pifq* mutant (University of California, Berkeley),
Prof. Haodong Chen (Peking University, China) for providing anti-PIF3 antibody, and Dr.
Jang In-Cheol (Temasek Life Sciences Laboratory, Singapore) for useful discussion.

## Author Contributions

**Conceptualization:** Pingzhi Zhao, Xuan Zhang, Xueping Zhou, Jian Ye.

**Funding acquisition:** Pingzhi Zhao, Xuan Zhang, Xueping Zhou, Jian Ye.

**Investigation:** Pingzhi Zhao, Xuan Zhang, Yuqing Gong, Duan Wang, Ning Wang, Yanwei
Sun, Lianbo Gao, Xueping Zhou.

**Methodology:** Dongqing Xu, Xing Wang Deng.

**Resources:** Dongqing Xu, Xing Wang Deng.

**Supervision:** Rong-Xiang Fang, Jian Ye.

**Writing – original draft:** Pingzhi Zhao, Xuan Zhang, Xueping Zhou, Jian Ye.

**Writing – review & editing:** Shu-Sheng Liu, Xing Wang Deng, Daniel J. Kliebenstein, Rong-Xiang Fang.

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
