## [Decision Letter · Decision Letter 0]

7 Aug 2020

Dear Dr. Ye,

Thank you very much for submitting your manuscript "Red-light is an environmental effector for mutualism between begomovirus and its vector whitefly" for consideration at PLOS Pathogens. As with all papers reviewed by the journal, your manuscript was reviewed by members of the editorial board and by several independent reviewers. In light of the reviews (below this email), we would like to invite the resubmission of a significantly-revised version that takes into account the reviewers' comments.

As reviewers point out one of the major issues that needs to be addressed is how PIFs integrate light signaling to attract whiteflies considering PIFs are degraded in light. Since PIFs are degraded in light, how could βC1 interact with PIFs? To demonstrate that βC1 does interact with PIFs, it will be required to perform the interaction studies using commercially available PIF antibodies in light and dark conditions. Furthermore, as reviewers point out, a more rigorous genetic relationship between PIFs, MYC2 and βC1 should be established to demonstrate that PIFs mediate plant defense against whiteflies and βC1 inhibit PIFs and MYC2..

We cannot make any decision about publication until we have seen the revised manuscript and your response to the reviewers' comments. Your revised manuscript is also likely to be sent to reviewers for further evaluation.

Sincerely,

Savithramma P. Dinesh-Kumar

Associate Editor

PLOS Pathogens

Shou-Wei Ding

Section Editor

PLOS Pathogens

Kasturi Haldar

Editor-in-Chief

PLOS Pathogens

orcid.org/0000-0001-5065-158X

Michael Malim

Editor-in-Chief

PLOS Pathogens

orcid.org/0000-0002-7699-2064

Reviewer's Responses to Questions

**Part I - Summary**

Reviewer #1: In this manuscript, the authors report red-light is an environmental effector for mutualism between begomovirus and its vector whitefly. Red-light is able to promote mutualism of whitefly-begomovirus through enhancing the stability of βC1 protein, which forms complex with PIFs. Functional analysis of PIFs indicates that PIFs positively modulate plant defense responses to whitefly via activating the transcription levels of TSP10. Furthermore, the exogenous MeJA assay combined with light assay confirmed that PIFs may integrates light and jasmonate signals through MYC2. Moreover, BiFC and transient assays showed that βC1 disrupts the homodimerization of PIFs and MYC2 and enhances the heterhomodimer of PIF-MYC2, therefore impairing plant defenses against begomovirus transmited by whitfly. Collectively, the authors try to demonstrate that how light signals involve in the mutualism between whitefly and its transmitted begomovirus.

The report is of interest to a broad audience and the topic is suitable for the PLoS Pathogens. The biochemical experiments were done in great detail and supported the conclusions in general.

Reviewer #2: In this manuscript, Zhao et al., reported that red-light is an environmental effector for mutualism between begomovirus and its vector whitefly. My main concern about this manuscript is that the light effect on host attraction to whitefly cannot be linked to the role of PHYTOCHROME-INTERACTING FACTORS (PIFs) transcription factors and its interaction with βC1. The evidence provided in this study is not sufficient to draw a conclusion that βC1 interacts and interferes with PIFs, and causes the light dependent host attraction to whitefly. In the dark, PIF proteins accumulate and directly regulate thousands of genes to maintain skotomorphogenesis. Upon illumination, the photoactived phytochromes trigger PIFs’ rapid phosphorylation and subsequently proteasome-mediated degradation. If PIFs link to the light effect on host attraction to whitefly, there should be no difference between light and dark treatment on host attraction to whitefly when using pifq (pif1/3/4/5) quadruple mutant. However, the data showed that pifq (pif1/3/4/5) quadruple mutant attracted more whitefly in white light but not in dark. How PIFs integrates light when the proteins were absent? If PIFs integrates light and regulate the transcription of terpene synthase genes, and if the above hypothesis work, in pifq (pif1/3/4/5) quadruple mutant, the expression of terpene synthase genes including AtTPS10、AtTPS14 and AtTPS21 should have no difference between dark and light treatment due to the absence of four PIFs. However, the expression of AtTPS10、AtTPS14 and AtTPS21 were downregulated in pifq (pif1/3/4/5) quadruple mutant in the dark, the expression of AtTPS21 was even upregulated in the mutant. Moreover, PIFs were degraded in white light or red light, and the host attraction to whitefly is caused by white or red light, how βC1 interacted with the PIF proteins that are not accumulated at all in light. If this is the case, the interaction of βC1 with the PIFs that obtained by yeast two hybrids and Co-IP experiments will not have any role on light signaling. The light effect may due to other mechanism but not interference of PIFs pathway by βC1.

Reviewer #3: In the MS, authors reported that red-light stabilized begomoviral βC1 protein. βC1 interacted with PHYTOCHROME-INTERACTING FACTORS (PIFs) transcription factors that positively controlled plant defenses against whitefly by directly binding to the promoter of terpene synthase genes for regulation. PIFs also interacted with MYC2 that also positively regulated the expression of terpene synthase genes, and the interaction interfered their regulation on TPS expression. Moreover, βC1 could strengthen such interaction as a bridge. Authors tried to focus on the function of red-light in the interaction among virus, vector and plant, which is a very interesting topic.

**Part II – Major Issues: Key Experiments Required for Acceptance**

Reviewer #1: However, in my opinion, it would be more rigorous if some genetic relationships among PIFs, MYC2 and βC1 will hold up.

1)Will PIF3 overexpression be able to recover the susceptible phenotype of βC1 overexpression plants against whitefly?

2)I would like to know the phenotype of βC1 overexpression in the background of pifq mutant? Which will further demonstrate the relationship between βC1 and PIFs against whitefly infestation.

3)How is the phenotype of pifq/myc2-1 feeding by whitefly? The authors demonstrated that PIFs integrates light and jasmonate signals by interaction with MYC2, and co-regulated the transcription of terpene synthase genes. However, the genetic relationship between PIFs and MYC2 is obscure.

Those experiments will clarify the relationships and strengthen the conclusions, and reveal more details between regulations of PIFs-βC1 and PIFs-MYC2.

Reviewer #2: Major concerns:

1. If PIFs integrates light and have the effects on whitefly attraction, there should be no difference between light and dark treatment on host attraction to whitefly when using pifq (pif1/3/4/5) quadruple mutant. How PIFs integrates light when the proteins were absent?

2. In the dark, PIF proteins accumulate and directly regulate thousands of genes to maintain skotomorphogenesis. Upon illumination, the photoactived phytochromes trigger PIFs’ rapid phosphorylation and subsequently proteasome-mediated degradation. In the dark, PIF proteins accumulated, and if PIFs regulate the transcription of terpene synthase genes, the expression of AtTPS10、AtTPS14 and AtTPS21 should be upregulated. But they were not (Figure 4E). In the light, PIF proteins were degraded, and if PIFs regulate the transcription of terpene synthase genes, the expression of AtTPS10、AtTPS14 and AtTPS21 should be downregulated in the wild type Col 0, but they were not, either (Figure 4E-G).

3. If PIFs regulate the transcription of terpene synthase genes and the expression of terpene synthase genes is light-dependent, the expression of AtTPS10、AtTPS14 and AtTPS21 should have no difference between light and dark treatment when using pifq (pif1/3/4/5) quadruple mutant in the dark. When PIFs were disrupted, How the mutants could still respond to light?

4. In the light, PIF proteins were degraded. The host attraction to whitefly is caused by white or red light. If the light effect is due to the interaction of βC1 with PIFs. How βC1 could interact with the PIF proteins that are not accumulated at all during the day light time.

5. In Fig 3A, we cannot see the results. This figure was not displayed normally.

6. Using BiFC assays to show the interaction between AtMYC2 with AtPIF3 and AtPIF4 (S6 Fig) are misleading, BiFC assays are easy to cause false positive results. The authors should provide the yeast two hybrid assay to confirm their interactions.

7. The finding that βC1 accumulated in light but not in dark is interesting. The light effect on host attraction to whitefly may due to other mechanism but not interference of PIFs pathway by βC1.

Reviewer #3: 1. In Fig 2 and S9, a control foreign protein should be detected to show that βC1 was specifically decreased in darkness. Also, surprisingly, fluorescence from YFP-fused βC1 in darkness was at a similar level to that in white light (S3 Fig). How to explain this?

2. In Fig 4, AtPIF3-OE was used for analysis. If βC1 was expressed in AtPIF3-OE, could the βC1-mediated whitefly preference be impaired?

3. In Fig 6C, βC1 increased the interaction signal of AtPIF4-cEYFP and nEYFP-AtMYC2. How do you exclude the roles of βC1 as suppressor of RNA silencing that could increase the expression level of both proteins? Moreover, in Fig 6E, with the increase of His-βC1, the amount of pulled down GST-AtMYC2 was not changed accordingly. Does it mean that βC1 only function as VSR improving the level of foreign protein?

**Part III – Minor Issues: Editorial and Data Presentation Modifications**

Reviewer #1: no

Reviewer #2: Minor concerns:

1. In Fig 5H, I strongly recommended to include a schematic diagram showing constructs for the effectors and the AtTPS10 Promoter : LUC reporter.

2. In abstract，Page 6，line 37 “inhibit” should be changed to “inhibits”.

3. In introduction，Page 6，line 50 “affect” should be changed to “affects”.

4. Fig 1D “��C1” should be changed to “�C1”.

5. In reference，Page 29，line 558 “Interactions” should be changed “interactions”; Page 32，line 676 “Interacting Factors” should be changed “interacting factors”.

Reviewer #3: 1. Results in this paper demonstrate the function of PIFs in the interaction among begomovirus, vector and plant, while the mechanism of red-light to stabilize βC1 is not clarified clearly. I suggest authors reorganizing the paper to focus on function of PIFs and its relationship to MYC2.

2. Line 86. Actually, the mechanism of how light affects this interaction is not clarified here either. It’s better to modify this sentence.

3. Line 90. Virus name here should not be italic. Please see https://talk.ictvonline.org/information/w/faq/386/how-to-write-virus-and-species-names

4. Line 112 and 124. “Nb” should be italic? Please check the full text.

5. In Fig 5A, do promoters of TPS14 and 21 have similar elements to that of TPS10? Are they also regulated by PIFs?

6. In Fig 5C, total YFP-fused proteins should be detected to ensure that proteins in GUS and βC1 treatment were at a similar level.

7. In Fig 6E, please provide additional evidences to confirm three proteins are in a complex. Also, it had better determine the responsible domains of βC1, PIF and MYC2 for interaction, which would help understand their interaction.

PLOS authors have the option to publish the peer review history of their article (what does this mean?). If published, this will include your full peer review and any attached files.

Reviewer #1: No

Reviewer #2: No

Reviewer #3: No
---

## [Decision Letter · Decision Letter 1]

31 Oct 2020

Dear Dr. Ye,

We are pleased to inform you that your manuscript 'Red-light is an environmental effector for mutualism between begomovirus and its vector whitefly' has been provisionally accepted for publication in PLOS Pathogens.

Best regards,

Savithramma P. Dinesh-Kumar

Associate Editor

PLOS Pathogens

Shou-Wei Ding

Section Editor

PLOS Pathogens

Kasturi Haldar

Editor-in-Chief

PLOS Pathogens

orcid.org/0000-0001-5065-158X

Michael Malim

Editor-in-Chief

PLOS Pathogens

orcid.org/0000-0002-7699-2064

Reviewer Comments (if any, and for reference):

Reviewer's Responses to Questions

**Part I - Summary**

Reviewer #1: According to the comments, the authors have conducted a series good experiments to further supporting how whitefly hijacks host external and internal signaling to enhance the mutualistic relationship with its insect vector. In my opinion, the revised manuscript is good for publishing at the PloS Pathogens.

Reviewer #2: The authors have now included new data showing that AtPIF3 proteins are detectable in light and AtPIF3 indeed interacts with βC1 in light and dark conditions. Most of questions have been addressed in the revised manuscript. The title of the manuscript is too big, it maybe narrow down to a more specific title.

Reviewer #3: (No Response)

**Part II – Major Issues: Key Experiments Required for Acceptance**

Reviewer #1: N/A

Reviewer #2: (No Response)

Reviewer #3: (No Response)

**Part III – Minor Issues: Editorial and Data Presentation Modifications**

Reviewer #1: N/A

Reviewer #2: (No Response)

Reviewer #3: (No Response)

PLOS authors have the option to publish the peer review history of their article (what does this mean?). If published, this will include your full peer review and any attached files.

Reviewer #1: No

Reviewer #2: No

Reviewer #3: No

---

## [Editor Report · Acceptance letter]

7 Jan 2021

Dear Dr. Ye,

We are delighted to inform you that your manuscript, "Red-light is an environmental effector for mutualism between begomovirus and its vector whitefly," has been formally accepted for publication in PLOS Pathogens.

Best regards,

Kasturi Haldar

Editor-in-Chief

PLOS Pathogens

orcid.org/0000-0001-5065-158X

Michael Malim

Editor-in-Chief

PLOS Pathogens

orcid.org/0000-0002-7699-2064